# Mitotane-Induced Hypothyroidism and Dyslipidemia in Adrenocortical Carcinoma: Sex Differences and Novel Evidence from a Thyroid Cell Model

**DOI:** 10.3390/curroncol32120700

**Published:** 2025-12-12

**Authors:** Irene Tizianel, Arianna Beber, Alberto Madinelli, Mario Caccese, Susi Barollo, Loris Bertazza, Elena Ruggiero, Simona Censi, Caterina Mian, Filippo Ceccato

**Affiliations:** 1Department of Medicine (DIMED), University of Padova, 35128 Padova, Italysusi.barollo@unipd.it (S.B.); loris.bertazza@unipd.it (L.B.); simona.censi@unipd.it (S.C.); caterina.mian@unipd.it (C.M.); 2Endocrinology Unit, University Hospital of Padova, 35128 Padova, Italy; 3Oncology Unit 1, Department of Oncology, Veneto Institute of Oncology IOV-IRCCS, 35128 Padova, Italy; mario.caccese@iov.veneto.it; 4Pain Therapy and Palliative Care Unit, Veneto Institute of Oncology IOV-IRCCS, 35128 Padova, Italy; elena.ruggiero@iov.veneto.it

**Keywords:** adrenocortical carcinoma, mitotane, central hypothyroidism, dyslipidemia, thyroid cell cytotoxicity, sex differences

## Abstract

Adrenocortical carcinoma (ACC) is a rare and aggressive malignancy treated with mitotane, a drug that often causes side effects like central hypothyroidism and dyslipidemia. This study found that mitotane directly damages thyroid cells, through laboratory experiments showing a dose-dependent toxic effect on thyroid tissue. Clinically, half of the patients developed central hypothyroidism, mostly women, while dyslipidemia affected mainly men. This new evidence of mitotane’s direct thyroid toxicity improves understanding of its side effects and highlights the importance of monitoring thyroid function during treatment.

## 1. Introduction

Adrenocortical carcinoma (ACC) is a rare but aggressive endocrine malignancy, characterized by poor prognosis and limited treatment options, especially in advanced stages. The estimated incidence of ACC ranges from 0.7 to 2 cases per million population per year [1]. Complete surgical resection remains the cornerstone of curative treatment; however, the high risk of recurrence, especially in patients with locally advanced disease, high-grade tumors, incomplete resection, or metastatic disease at diagnosis, often necessitates adjuvant therapies [2].

Mitotane, an adrenolytic agent derived from the insecticide DDT, is the only approved systemic therapy for ACC and has been widely used in both adjuvant and palliative settings. In high-risk patients, mitotane has been shown to prolong recurrence-free survival following surgery [3,4]. Current guidelines recommend at least 2 years (and up to 5 years) of adjuvant mitotane treatment in patients with standard/high risk of recurrence [1,2].

Beyond its cytotoxic effects on adrenocortical tissue, mitotane acts as a potent inhibitor of steroidogenesis and exerts endocrine and metabolic effects. Common mitotane-induced toxicities include adrenal insufficiency, hypogonadism, dyslipidemia, and hypothyroidism [5,6]. Among these, mitotane-induced hypothyroidism is particularly noteworthy due to its central (secondary) origin, likely resulting from mitotane-induced dysfunction of the hypothalamic–pituitary axis [7]. Although it has been reported in several case series and retrospective studies, the pathophysiological mechanisms remain incompletely understood. Proposed hypotheses include direct suppression of hypothalamic-pituitary function, increased peripheral metabolism of thyroid hormones, or alterations in thyroid hormone-binding proteins. Importantly, mitotane treatment is also known to induce significant changes in lipid metabolism, often leading to marked hypercholesterolemia and hypertriglyceridemia, further complicating the clinical picture [8].

Dyslipidemia is a particularly prominent and early-onset metabolic disturbance associated with mitotane therapy. Patients commonly develop significant hypercholesterolemia, often with marked elevation of low-density lipoprotein cholesterol (LDL-c), and hypertriglyceridemia, which are believed to result from both increased hepatic synthesis and altered lipid clearance mechanisms. The proposed mechanisms include mitotane-induced activation of hepatic nuclear receptors such as SREBP and LXR, in addition to altered thyroid hormone metabolism and direct mitochondrial toxicity. Importantly, dyslipidemia can be severe and persistent, often requiring pharmacologic lipid-lowering therapy, and may carry long-term cardiovascular implications [9]. The efficacy of statins may be limited by mitotane-induced CYP3A4 induction, which accelerates drug metabolism [5,9]. For this reason, current guidelines recommend pravastatin or rosuvastatin as preferred agents, although clinical data remain scarce [1]. Monitoring for mitotane-induced adverse effects (physical and laboratory examination) is recommended every 3–4 weeks during the initial phase and every 2–3 months during follow-up. Given the scarcity of prospective clinical data, a better understanding of the endocrine and metabolic side effects of mitotane is crucial for optimizing patient management.

This study aimed to evaluate the incidence and clinical features of mitotane-induced central hypothyroidism and dyslipidemia in patients with ACC, and to assess their potential relationship. Complementary in vitro experiments using the FRTL-5 rat thyroid cell line were conducted to investigate the direct, dose-dependent effects of mitotane on thyroid cell viability.

## 2. Materials and Methods

### 2.1. Study Design

We conducted a retrospective cohort study on our cohort of ACC patients followed at our Endocrinology Unit in Padova University-Hospital from 2005 to 2025. Approved by local Ethics Committee: (protocol number 22546, registration 2023-120). Data are available in the Repository of the University of Padova.

### 2.2. Patient Selection and Cell-Based Analysis

The inclusion criteria were age ≥18 years, histologically confirmed ACC diagnosis, exposure to adjuvant or palliative mitotane for at least 6 months, availability of thyroid hormone/lipid profiles data, and their respective pharmacological treatments. Patients with incomplete or unavailable data were excluded. Clinical data were extracted from electronic medical records at approximately three-month intervals, starting from the ACC diagnosis until the last available follow-up. Baseline data were referred to the ACC diagnosis, before mitotane start. Plasma mitotane concentration was measured by the Lysosafe service provided by HRA Pharma—ESTEVE (www.lysosafe.com).

Plasma mitotane concentration was defined as target, low, or high according to the reference range (14–20 mg/L). To better assess mitotane exposure over time, we also calculated the time in target range (TTR), defined as the number of months in which mitotane plasmatic levels were within the reference range [4]. Thyroid function (TSH, free T4, free T3) and lipid profile (total cholesterol, HDL, triglycerides) were assessed periodically starting from ACC diagnosis, before the initiation of mitotane therapy, and afterwards at each follow-up visit every 3–4 months, accordingly to local clinical practice. Central hypothyroidism was diagnosed based on European Thyroid Association guidelines using the following criteria: low free T4 associated with low or normal TSH values, in at least two repeated measurements [10].

Dyslipidemia was defined according to ESC/EAS Guidelines for the management of dyslipidemia [11], or in case of current pharmacological treatment. Medical treatment of hypothyroidism and dyslipidemia was assessed. We distinguished pre-existing dyslipidemia from mitotane-induced dyslipidemia by reviewing baseline lipid profiles prior to mitotane initiation.

To ensure greater accuracy, the pharmacological management of lipid disorders was based on the guideline recommendations available at that time and subsequently standardized using a stepwise approach, involving multiple treatment stages (Step 0–4). Steps were based on predefined criteria considering the therapeutic equivalence among different active agents (statins, ezetimibe, and bempedoic acid), as shown in Table A1 [12].

The rat thyroid cell line FRTL-5 was used as a model to investigate normal thyroid. MTT assay was performed to evaluate the effect of increasing mitotane concentrations on FRTL-5 cell viability. The analyzed range was from 10 to 100 µM, corresponding to mitotane plasma levels of 3.2–32 mg/L, respectively; therefore, 50 µM corresponds to 16 mg/L of mitotane plasma levels, as previously reported by Zatelli et al. in 2010 [7]. In our experiment, cells were plated on 96-well tissue-culture microtiter plates at a density of 5 × 10^3^ cells per well. The next day, the cells were treated with mitotane for 24 or 48 h and both in the presence and absence of TSH. Untreated cells (with and without TSH) were used as negative controls, and for each treatment timepoint, the cell viability of the mitotane doses was compared to their corresponding untreated control. We measured the viability effect on cell viability using 3-(4,5-dimethylthiazol-2-yl)-2,5 diphenyltetrazolium bromide (MTT) (Sigma-Aldrich, St. Louis, MO, USA) [13,14]. Then the culture media were removed, and 100 µL DMSO was added to dissolve the formazan crystals, and absorbance was measured at 570 nm (Viktor 3, Perkin-Elmer, Waltham, MA, USA). The experiments were repeated three times, each containing three technical replicates of each measured point.

Parallelly, to confirm the viability data and to overcome eventual limitations of the MTT assay, Incucyte^®^ S3 Live-Cell Analysis (Essen Bioscience, Welwyn Garden City, Hertfordshire, UK) was used to evaluate the same conditions and mitotane ranges of concentration.

In our experiments, cells were seeded in a sterile 96-well plate (10,000 cells/well) in 100 µL of complete culture medium and incubated overnight at 37 °C and 5% CO_2_. The day after, cells were treated with mitotane, and proliferation was monitored for 48 h using Incucyte^®^ S3 Live-Cell Analysis Instrument (10× objective). Cell confluence was recorded every 6 h by phase-contrast scanning for 2 days at 37 °C and 5% CO_2_ and calculated from the microscopy images. Images were analyzed using the Incucyte software (GUI Version 2022A, details 20221.1.80.81.399994). The experiments were repeated twice, and for each, four different fields of each treated well on the plate were photographed. Since three technical replicates were performed for each treatment, a total of 12 measurements were available for each point.

### 2.3. Statistical Analysis

Categorical variables were reported as frequencies and percentages, while continuous ones were reported as median and interquartile range (IQR). Nonparametric tests were used to compare groups. Fisher’s exact test, with Bonferroni correction for multiple comparisons, was used to compare proportions between groups. Mann–Whitney U-test, or Kruskal–Wallis test, was used for comparison of continuous variables between two or three independent groups, respectively. Time to central hypothyroidism and dyslipidemia’s onset was calculated using the Kaplan–Meier method, and comparisons between groups were performed by the log-rank test. Cox proportional hazards regression, univariate, and multivariate analyses were performed to identify factors that could potentially influence central hypothyroidism and dyslipidemia development. Hazard ratio (HR) and 95% confidence interval (CI) were evaluated. Data analysis was performed using SPSS version 24 software package for Windows (SPSS Inc., Version 24, Chicago, IL, USA). Statistical significance was accepted at *p* < 0.05.

## 3. Results

Our final cohort, depicted in Figure A1, was made up of 38 ACC patients; at the last available follow-up, 28 (74%) patients were alive, and 10 (26%) patients were dead. Median follow-up time after mitotane start was 25 months (11.25–40 months).

The median age at diagnosis was 55.5 years (range: 18–79 years), and 28 (74%) patients were female. According to ENSAT staging, 1, 5, 13, and 19 patients were classified as stage I, II, III, and IV, respectively. Fifteen patients (44.1%) had non-secreting tumors, 11 (32.4%) had cortisol-secreting tumors, 7 (18.4%) had mixed cortisol and androgen secretion, and 1 patient had androgen-only secretion. All patients received mitotane for at least 6 months, alone (n = 9) or combined with chemotherapy (n = 29) with etoposide, doxorubicin, and cisplatin (EDP-M) according to ENSAT staging, resection status, and recurrence risk.

### 3.1. Central Hypothyroidism

At ACC diagnosis, before starting mitotane, 4/38 patients had known primary autoimmune hypothyroidism, and the remaining 34 had normal thyroid function. During follow-up, central hypothyroidism developed in 17/34 (50%) of these patients, with the remaining 17 maintaining normal thyroid function during follow-up (median follow-up 20 months [IQR 6.5–42.5]). The median time to onset of central hypothyroidism from mitotane initiation was 25 months [IQR 0.82–49.18 months]; 75% of cases occurred within 40 months. Among the 17 affected patients, 15 were female, and 2 were male (88 vs. 12%, *p* = 0.024).

Levothyroxine (LT4) treatment was started upon detection of reduced FT4 levels with the aim of restoring to FT4 in the normal range. LT4 dosing data were available for 15 out of 17 patients with central hypothyroidism. The median LT4 dose was 0.91 mcg/kg/day (IQR 0.73–1.14). 9 out of 17 patients with mitotane-induced central hypothyroidism also underwent chemotherapy with EDP. Kaplan–Meier analysis showed a significant difference in hypothyroidism onset by sex (Log-rank χ^2^ = 8.460, *p* = 0.004), with female patients at higher risk, as shown in Figure 1.

Survival analysis revealed no significant difference in overall survival between patients with or without central hypothyroidism, considering the entire cohort: ENSAT stage I-IV (*p* = 0.894). This finding was consistent in the subgroup of patients ENSAT stage III–IV (*p* = 0.769).

At the onset time of central hypothyroidism, mitotane levels were in target (14–20 mg/L) in 8 patients, above target (>20 mg/L) in 3 patients, and below target (<14 mg/L) in 6 patients. To enhance accuracy, time in target range (TTR) from mitotane start to the last follow-up visit was assessed.

Univariate Cox regression identified female sex as a significant predictor of central hypothyroidism (HR = 7.73; 95% CI: 1.63–36.64; *p* = 0.010). In addition, a longer duration of mitotane time in the target range (TTR) was associated with an increased risk of developing central hypothyroidism (HR = 1.02 per month; 95% CI: 1.002–1.043; *p* = 0.034). Age at diagnosis (HR = 0.99; *p* = 0.657) and hormonal secretion (HR = 2.63; *p* = 0.092) were not significantly associated with the outcome (Table 1).

In multivariate Cox analysis adjusting for sex and TTR, female sex remained significantly associated with hypothyroidism onset (HR = 6.41, 95% CI: 1.36–30.2, *p* = 0.019), whereas TTR was no longer statistically significant (HR = 1.01, *p* = 0.102), as shown in Table 2.

### 3.2. Effect of Mitotane on FRTL-5 Cells Viability

MTT assay was conducted to assess the half maximal inhibitory concentration (IC50) of mitotane on FRTL-5 cells inhibition, from 24 to 48 h. Our data showed that IC50 was approximately 50 µM, with no significant difference after 24 to 48 h of cell incubation, and also in the presence or absence of TSH in the cell culture (Figure 2A–D). At concentrations greater than 50 µM, mitotane demonstrated a strong effect on FRTL-5 cells inhibition, with a maximal effect at 100 µM, corresponding to complete cell inhibition.

MTT assay results were confirmed by Incucyte^®^ S3 Live-Cell Analysis Instrument: at a mitotane concentration of 50 µM, we can observe that FRTL-5 area confluence remained stable over time. (Figure 2E,F and Figure 3).

### 3.3. Dyslipidemia

At diagnosis, before starting mitotane treatment, 25/38 (67%%) patients had normal lipid metabolism, 12/38 had known hypercholesterolemia, and 1 patient had combined hypercholesterolemia and hypertriglyceridemia. Excluding patients with pre-existing lipid abnormalities, 10 out of 25 (40%) patients developed mitotane-induced dyslipidemia during follow-up. Dyslipidemia occurred more frequently in males, 5/7 (71%), than in females, 5/18 (28%). Chi-square analysis indicated a statistically significant association between sex and dyslipidemia (χ^2^ = 4.001, *p* = 0.045).

Median time to dyslipidemia onset from mitotane start was 20 months (95% CI: 6.2–33.8 months). According to gender, the median time to dyslipidemia was 6 months in males (95% CI: 2.2–9.8 months) and 20 months in females (95% CI: 10.4–29.6 months).

Dyslipidemia was managed pharmacologically with statin, preferably rosuvastatin, eventually combined with ezetimibe, considering the recommendation to use statins not metabolized by CYP3A4 [1]. Treatment was standardized according to a stepwise protocol (STEP 0–4). Different STEPS are reported in Appendix A Table A1. Figure 4 illustrates the STEP distribution over time, assessed at each follow-up visit (every 3 months), and stratified into two groups: Group A (patients with pre-existing dyslipidemia before starting mitotane) and Group B, patients with mitotane-induced dyslipidemia. In Group B, treatment intensity progressively increased during mitotane therapy. Statistically significant differences were observed when comparing the 3-month evaluation with later timepoints at 12, 18, 24, and 30 months (*p* = 0.044, *p* = 0.026, *p* = 0.025, and *p* = 0.024, respectively). This analysis was not performed in Group A due to incomplete longitudinal data.

Kaplan–Meier analysis of time to onset of dyslipidemia stratified by gender revealed a statistically significant difference in the dyslipidemia-free survival distributions between the two groups (χ^2^ = 3.761, *p* = 0.04), indicating that the risk of developing dyslipidemia over time differs significantly by sex, with male patients at higher risk (Figure 5).

In univariate Cox regression analysis, female sex was significantly associated with a lower risk of developing dyslipidemia (HR = 0.16, 95% CI: 0.03–0.93, *p* = 0.042). No other clinical variable, including age, hormonal secretion, or mitotane time in target range (TTR), was significantly associated with dyslipidemia risk (Table 3). No cardiovascular events were reported during the follow-up period.

### 3.4. Relationship Between Mitotane-Induced Central Hypothyroidism and Dyslipidemia

Among the 17 patients who developed mitotane-induced central hypothyroidism, 7/17 (41%) also developed mitotane-induced dyslipidemia; however, this incidence did not differ among patients with no central hypothyroidism 10/17 (41 vs. 59%, respectively, *p* = 0.247). Cox proportional hazards regression was used to evaluate the relationship between mitotane-induced central hypothyroidism and the risk of dyslipidemia (Table 3). Patients with central hypothyroidism showed a non-significant trend toward increased dyslipidemia risk (HR = 2.01, 95% CI: 0.71–5.68, *p* = 0.188). The Kaplan–Meier analysis showed no significant difference in dyslipidemia-free survival between patients with and without central hypothyroidism (39.8 vs. 41.5 months, respectively, *p* = 0.344).

### 3.5. Lipid Profile Evolution After Mitotane Discontinuation

Four patients in our cohort discontinued mitotane treatment after 2 to 5 years, according to their free-of-disease state. Post-discontinuation lipid profile data were available for three of them at 3–4 months after treatment withdrawal, and are presented in Figure 6.

## 4. Discussion

### 4.1. Central Hypothyroidism

In our cohort, 50% of patients (17/34) with normal baseline thyroid function developed central hypothyroidism during mitotane therapy, with a median time to onset of 25 months and a higher prevalence in females. Female sex was a significant independent predictor, consistent with multivariate analysis. To our knowledge, no previous studies have specifically identified female sex as a clinical predictor of central hypothyroidism in patients treated with mitotane. Therefore, our findings may provide new insights into the potential role of sex-based susceptibility in the development of this endocrine toxicity.

Interestingly, central hypothyroidism occurred across a wide range of mitotane plasma levels, including below, within, and above the therapeutic window, suggesting that cumulative exposure assessed by TTR and individual susceptibility may be more relevant than single timepoint plasma levels.

A large multicenter retrospective study reported a higher prevalence of central hypothyroidism [15]: 95.5% among mitotane-treated ACC patients, with most cases developing within the first year: 33.3% occurred <3 months, 19.1% at 3–6 months, 14.3% at 6–9 months, and 9.5% at 9–12 months; at least 14.3% occurred after 12 months. Partial or complete recovery occurred in 65.4% of patients within the first two years following treatment discontinuation. A recent systematic review involving 493 patients (including pediatric and adult cases) found that FT4 reduction occurred in 45.4% of cases within 3–6 months of treatment initiation, supporting the notion that TSH is an unreliable marker of thyroid function in this context [9].

The underlying pathophysiology remains incompletely understood. Proposed mechanisms include suppression of thyrotroph cell function, altered TSH glycosylation impairing bioactivity, reduced TRH responsiveness, enhanced deiodinase activity (leading to increased FT3-to-FT4 ratio), and modifications in thyroid hormone-binding protein levels. Importantly, TSH values may be misleading due to a combination of biochemical and pharmacodynamic changes, making FT4 the most reliable marker for diagnosis and monitoring [9].

According to the 2018 European Thyroid Association guidelines, central hypothyroidism should be diagnosed based on low FT4 with inappropriately low or normal TSH. In line with this, recent reviews strongly recommend FT4-guided monitoring every 3–4 months during mitotane therapy [10]. This approach was adopted in our cohort, with levothyroxine initiated regardless of clinical symptoms, and normalization of FT4 was achieved within 1–3 months of supplementation. As patients in our cohort remained on levothyroxine treatment, longer follow-up will be required to assess thyroid function recovery following mitotane withdrawal. Despite its frequency, central hypothyroidism did not impact overall survival in our analysis, which aligns with previous studies suggesting that endocrine side effects of mitotane, while clinically significant, do not necessarily influence oncological prognosis.

In our study, we also analyzed the effect of increasing mitotane concentrations on cell viability in a rat thyroid cell line (FRTL-5). Regarding the MTT assay, our data showed that IC50 was approximately 50 µM, corresponding to a mitotane plasma concentration of 16 mg/L (therapeutic range 14–20 mg/L), as previously reported by Zatelli et al. [7], with no significant difference after 24 to 48 h of cell incubation and in the presence or absence of TSH in the cell culture. At concentrations greater than 50 µM, mitotane demonstrated a strong effect on FRTL-5 cell inhibition, with a maximal effect at 100 µM, corresponding to a complete cell inhibition. In 2010, Zatelli et al. demonstrated a mitotane direct effect on thyrotrope cell viability in a mouse pituitary cell line (TαT1), due to apoptosis activation and direct toxicity. Concerning apoptosis, mitotane induced a dose-dependent activation in caspase 3/7 activity, as previously reported [7]. Viability was assessed in TαT1 cells, after incubation with increasing mitotane concentrations (10–100 µM, corresponding to mitotane plasma levels of 3.2–32 mg/L). After a 48 h incubation, cell viability was significantly reduced due to mitotane treatment at 60 µM (−31%; *p* < 0.05), 80 µM (−53%; *p* < 0.01), and 100 µM (−75.5%; *p* < 0.01) compared with vehicle-treated control cells [7]. These findings allow us to make interesting considerations also in relation to our results.

Indeed, our experiment on FRTL-5 cells viability showed that 50% of FRTL-5 cells were inhibited by a mitotane concentration of 50 µM after 48 h. These results, compared to the 80 µM concentration needed to reduce cell viability of −53% in TαT1 cells, seem to imply greater thyroid toxicity, with respect to pituitary, for the same mitotane concentration. However, this conclusion is limited by the differences in cell models and experimental setups, and should be considered preliminary.

Therefore, it appears that mitotane-induced hypothyroidism has not only a pituitary implication, mimicking a central hypothyroidism, but also a direct primary mechanism, due to thyroid toxicity. Clinically, we observe central hypothyroidism, as TSH levels do not rise in response to low FT4. However, it cannot be excluded that a concurrent primary thyroid damage induced by mitotane causes thyroid cell dysfunction, resulting in overlapping primary hypothyroidism. Other conditions can further reduce TSH levels, such as high doses of glucocorticoids used as replacement therapy during mitotane treatment or TSH inactivation through abnormal glycosylation induced by mitotane [16,17]. Future studies could investigate the effects of mitotane on thyroid function through measurement of thyroid hormones and thyroglobulin in the supernatant of thyroid cell cultures. Such analyses would help to directly link in vitro cytotoxic effects to the clinical thyroid dysfunction observed in patients treated with mitotane.

### 4.2. Dyslipidemia

Mitotane-induced dyslipidemia occurred in 40% of patients with no prior history of lipid disorders, more frequently in males (71% vs. 28% in females), with earlier onset in men (median: 6 vs. 20 months). Male sex independently predicted dyslipidemia risk, as previously reported.

These findings are supported by previous reports showing that mitotane therapy leads to lipid abnormalities in 49–54% of patients, particularly elevation of LDL-C, HDL-C, and, in some cases, triglycerides [9]. One retrospective study demonstrated a mean increase of ~46 mg/dL in LDL-C and ~33 mg/dL in HDL-C, with peak levels observed after ~6–8 months of treatment. Notably, lipid abnormalities were more pronounced and persistent in male patients [18].

The mechanisms behind mitotane-induced dyslipidemia are multifactorial and include increased hepatic cholesterol synthesis via stimulation of HMG-CoA reductase; induction of hepatic cytochrome P450 enzymes, particularly CYP3A4, affecting HDL metabolism; impaired lipid clearance due to SR-B1 receptor modulation; inhibition of CYP11A1, reducing cholesterol conversion to pregnenolone; indirect effects of mitotane-induced hypothyroidism, which itself promotes hypercholesterolemia. Current recommendations advise regular lipid monitoring and pharmacological intervention when indicated. The induction of CYP3A4 by mitotane can reduce the efficacy of statins metabolized by this pathway (e.g., simvastatin, atorvastatin); hence, pravastatin or rosuvastatin are preferred agents [9].

In our study, most patients were managed using a standardized stepwise lipid-lowering protocol, with increasing intensity over time, particularly in those who developed mitotane-induced dyslipidemia. Rosuvastatin, alone or in combination with ezetimibe, was the most frequently used regimen. Despite effective lipid-lowering therapy, no cardiovascular events were recorded during follow-up. However, the long-term consequences of sustained dyslipidemia in this population remain unclear and warrant prospective evaluation. The observed sex differences in toxicity profiles may reflect biological differences in metabolism, drug clearance, or hormone sensitivity and represent a novel area for future investigation.

### 4.3. Relationship Between Mitotane-Induced Central Hypothyroidism and Dyslipidemia

Dyslipidemia in hypothyroidism is driven by altered lipid metabolism due to reduced thyroid hormones and increased TSH levels. Recent studies highlight additional factors like PCSK9 and ANGPTL proteins that contribute to lipid abnormalities and impaired HDL function. These mechanisms help explain the complex dyslipidemia observed in patients treated with mitotane, who often develop hypothyroidism as a side effect, underscoring the need for careful lipid monitoring and management [19]. Future prospective studies are warranted to assess the impact of levothyroxine therapy on the potential improvement of dyslipidemia in patients treated with mitotane.

Our findings are in line with a larger retrospective series of 74 ACC patients, where Basile et al. [5] reported a 36.2% prevalence of hypothyroidism and a 50% rate of statin initiation for significant dyslipidemia, which emerged after a median of 6 months of mitotane treatment. Importantly, most endocrine and metabolic toxicities appeared even before plasma mitotane concentrations reached therapeutic levels, emphasizing the need for vigilant early monitoring [20,21].

A complementary retrospective study conducted in 50 Danish ACC patients [8] further illustrates that mitotane induces significant metabolic and hormonal alterations. After six months of therapy, total cholesterol rose markedly (from a median 5.1 to 7.4 mmol/L, *p* < 0.001), accompanied by increases in LDL, HDL, and triglycerides (all *p* ≤ 0.03). Thyroxine (total and free T4) levels significantly declined on treatment while TSH remained stable (both *p* < 0.001), underscoring the unreliability of TSH as a monitoring marker. Importantly, administration of statins for three months reduced total and LDL cholesterol, and mitotane discontinuation led to further lipid improvements.

### 4.4. Limitations

Limitations of this study include its retrospective design, relatively small sample size, and the absence of quality-of-life assessment tools. Additionally, the cohort may be affected by survivorship bias, as only patients who survived at least six months after starting mitotane were included. Patients with more aggressive disease who died early might be underrepresented, and their endocrine outcomes remain unknown. Furthermore, data on the effectiveness of levothyroxine therapy in improving dyslipidemia are currently insufficient. Finally, given the relatively small number of events, some degree of overfitting cannot be entirely excluded, although we limited the number of covariates and performed sensitivity analyses to mitigate this risk. However cautious interpretation of these findings is required, and they should be confirmed in larger studies.

## 5. Conclusions

Our study confirms that central hypothyroidism and dyslipidemia are frequent, clinically relevant, but manageable adverse effects of mitotane therapy in ACC. While these side effects did not influence overall survival, they require systematic monitoring and management to prevent complications and preserve quality of life. Despite a demonstrated direct effect on thyrotrope cell viability, our study firstly evidenced a direct thyroid toxicity of increasing mitotane concentrations on a rat thyroid cell line (FRTL-5).

## Figures and Tables

**Figure 1 curroncol-32-00700-f001:**
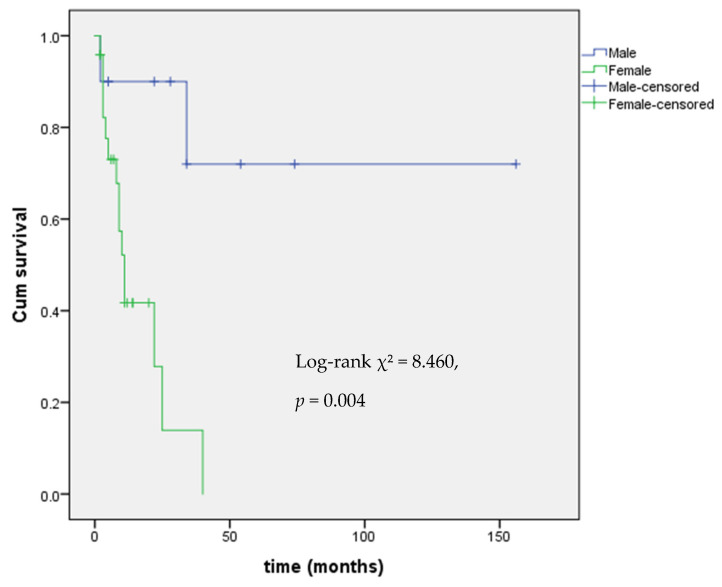
Kaplan–Meier analysis of event-free survival from central hypothyroidism by gender.

**Figure 2 curroncol-32-00700-f002:**
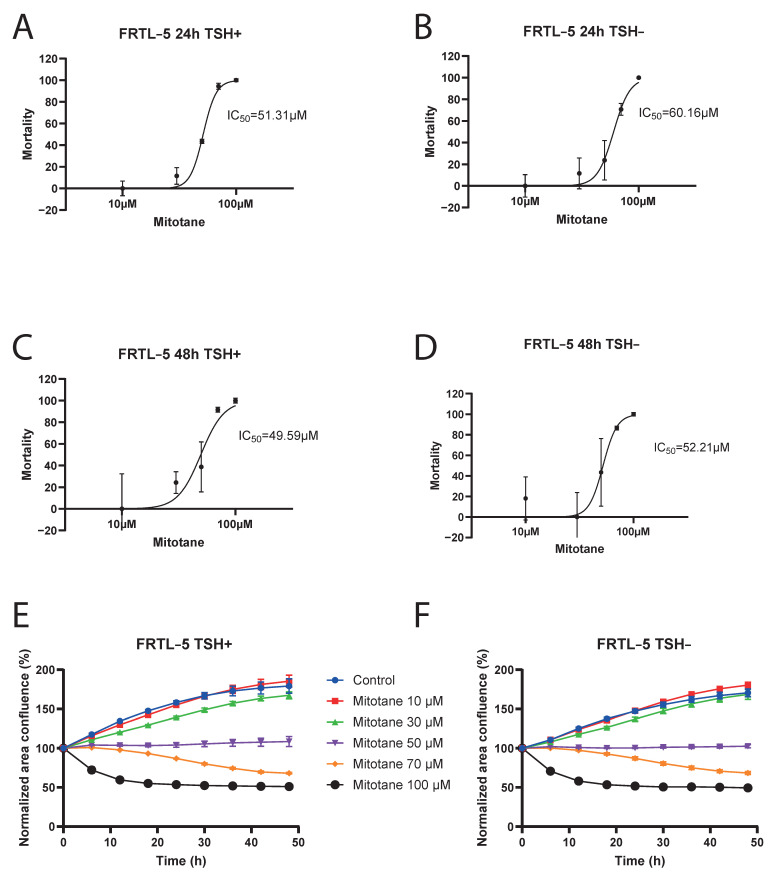
Logarithmic-dose vs. response graph (MTT assay FRTL-5). Panel (**A**) 24 h TSH+. Panel (**B**) 24 h TSH−. Panel (**C**) 48 h TSH+. Panel (**D**) 48 h TSH−. Incucyte^®^ S3 Live-Cell Analysis Instrument results: Panel (**E**) FRTL5 TSH+. Panel (**F**) FRTL5 TSH−.

**Figure 3 curroncol-32-00700-f003:**
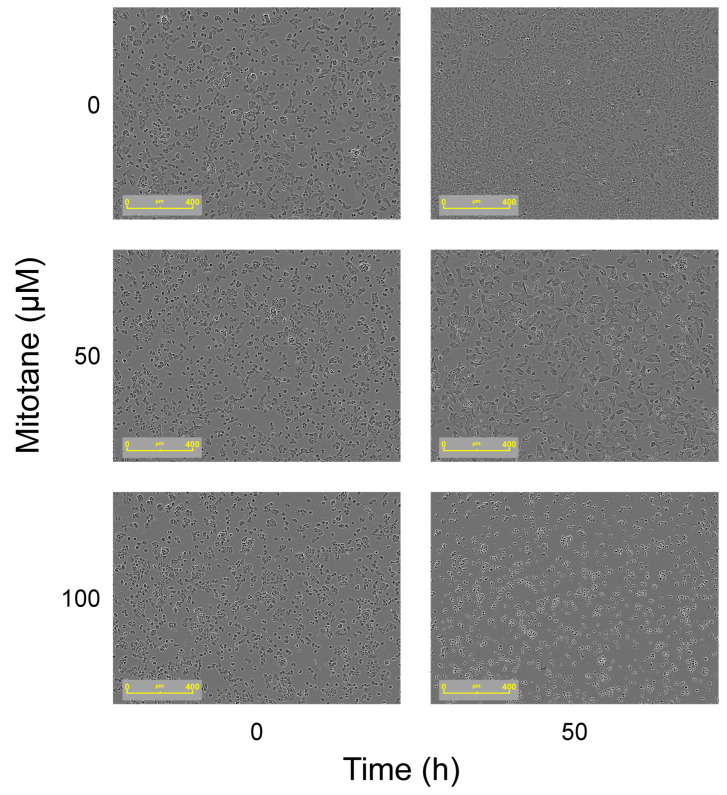
Effect on cell morphology over time (time 0 and 50 h) with increasing doses of mitotane (50 and 100 µM). As depicted in the figure on the right, the growth of cells was inhibited after 50 h with both 50 and 100 µM of mitotane. Images acquired using the Incucyte S3 Live-Cell Analysis System.

**Figure 4 curroncol-32-00700-f004:**
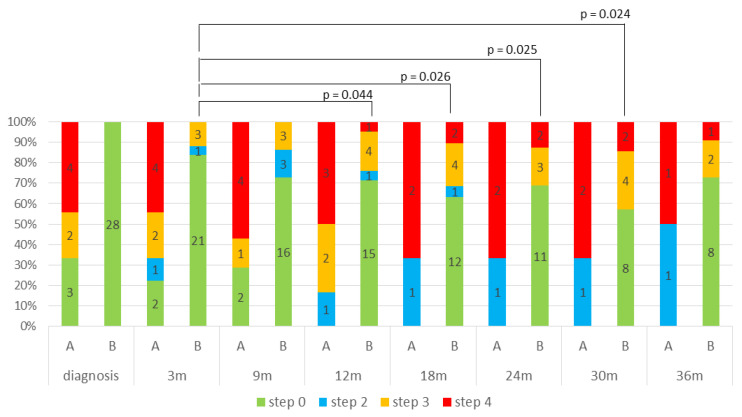
STEP distribution over time, stratified into two groups: Group A (patients with pre-existing dyslipidemia before starting mitotane) and Group B, patients with mitotane-induced dyslipidemia.

**Figure 5 curroncol-32-00700-f005:**
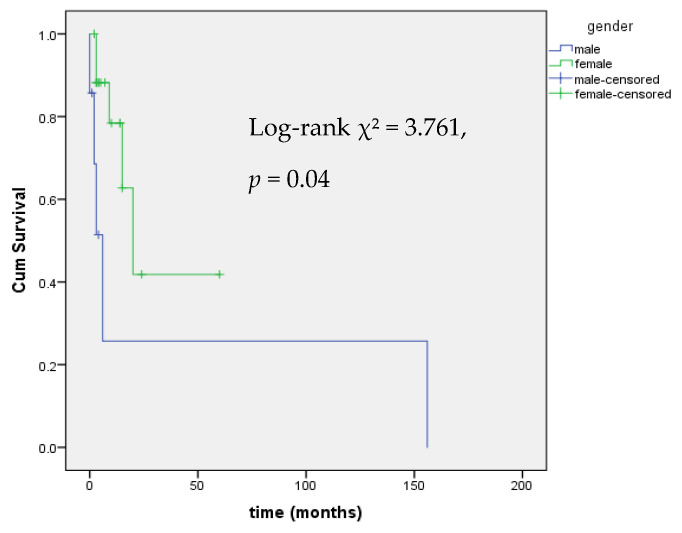
Kaplan–Meier Analysis of event-free survival from mitotane-induced dyslipidemia by gender.

**Figure 6 curroncol-32-00700-f006:**
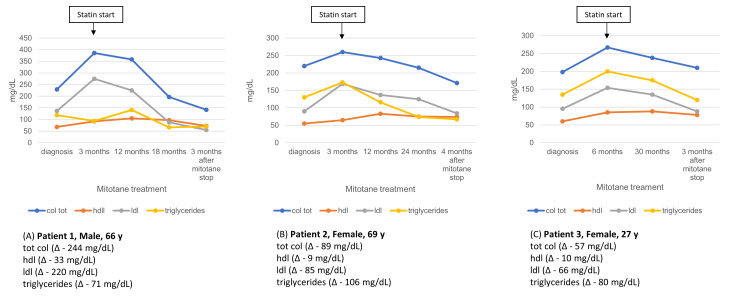
Longitudinal lipid profile in three patients during mitotane therapy and after statin initiation (arrow). Values are shown from diagnosis through follow-up and 3 months post-mitotane discontinuation, with absolute lipid changes (Δ) reported.

**Table 1 curroncol-32-00700-t001:** Univariate Cox regression analysis of clinical variables associated with the risk of central hypothyroidism.

	HR (95% CI)	*p* Value
Sex (F vs. M)	7.73 (1.63–36.64)	0.010
Age at diagnosis (years)	0.99 (0.96–1.03)	0.657
Hormonal secretion (cortisol/androgen secreting vs. non-secreting)	2.63 (0.85–8.11)	0.092
TTR months (%)	1.02 (1.002–1.043)	0.034

Abbreviations: TTR: mitotane time in target range.

**Table 2 curroncol-32-00700-t002:** Multivariate Cox regression analysis of clinical variables associated with the risk of central hypothyroidism.

	HR (95% CI)	*p* Value
Sex (F vs. M)	6.41 (1.36–30.2)	0.019
TTR months (%)	1.01 (1.002–1.043)	0.102

Abbreviations: TTR: mitotane time in target range.

**Table 3 curroncol-32-00700-t003:** Univariate Cox regression analysis of clinical variables associated with the risk of dyslipidemia.

	HR (95% CI)	*p* Value
Sex (F vs. M)	0.16 (0.03–0.93)	0.042
Age at diagnosis (years)	0.99 (0.94–1.05)	0.854
Hormonal secretion (non-secreting vs. cortisol/androgen secreting)	0.96 (0.22–4.06)	0.960
TTR months (%)	0.99 (0.96–1.02)	0.842
Central hypothyroidism (yes vs. no)	2.01 (0.75–5.96)	0.188

## Data Availability

Data are available upon request to the corresponding author.

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
