# Peer review of "Mitotane-Induced Hypothyroidism and Dyslipidemia in Adrenocortical Carcinoma: Sex Differences and Novel Evidence from a Thyroid Cell Model"

_curroncol, 2025, doi:10.3390/curroncol32120700_

Round 1
Reviewer 1 Report
Comments and Suggestions for Authors
In the manuscript titled “Mitotane-induced hypothyroidism and dyslipidemia in adrenocortical carcinoma: sex differences and novel evidence from a thyroid cell model,” the authors investigate the development of central hypothyroidism and dyslipidemia in a cohort of 38 ACC patients, identifying sex differences in their occurrence.
Although the study is interesting, from my perspective the main limitation is that it appears to be centered on mitotane, yet only 9 out of 38 patients received mitotane alone, while the majority (29/38) were treated with a combination of mitotane and chemotherapy. Therefore, my concerns are the following:
1. Chemotherapy (EDP) can also induce central hypothyroidism and dyslipidemia. For this reason, the authors should compare the incidence of hypothyroidism and dyslipidemia in patients treated with mitotane alone vs those treated with EDPM. I understand that the sample size is limited; however, the authors could at least strengthen the in vitro section by comparing the cytotoxic effects of the individual EDP components in the FRTL-5 cell line.
2. The doubling time of the FRTL-5 cell line is approximately 36–38 hours, and it is generally preferable to treat cells for a duration longer than their doubling time. Why did the authors choose 24 and 48 hours as treatment durations? I suggest using 48 and 96 hours instead. Additionally, the standard deviations in the graphs are quite high, making interpretation of the results challenging. How many times were the experiments repeated? Please specify.
3.To increase the relevance of the in vitro component, it would also be helpful to compare the effects of mitotane and EDP in an ACC cell line such as NCI-H295R.
4. To better connect the treatments to hypothyroidism, molecular analyses are needed. For explample, as the authors mention in the discussion, it would be valuable to explore the effect of mitotane on thyroid function through the measurement of thyroid hormones and thyroglobulin in the supernatant of thyroid cell cultures.
Author Response
Reviewer #1
In the manuscript titled “Mitotane-induced hypothyroidism and dyslipidemia in adrenocortical carcinoma: sex differences and novel evidence from a thyroid cell model,” the authors investigate the development of central hypothyroidism and dyslipidemia in a cohort of 38 ACC patients, identifying sex differences in their occurrence.
Although the study is interesting, from my perspective the main limitation is that it appears to be centered on mitotane, yet only 9 out of 38 patients received mitotane alone, while the majority (29/38) were treated with a combination of mitotane and chemotherapy. Therefore, my concerns are the following:
Question 1. Chemotherapy (EDP) can also induce central hypothyroidism and dyslipidemia. For this reason, the authors should compare the incidence of hypothyroidism and dyslipidemia in patients treated with mitotane alone vs those treated with EDPM. I understand that the sample size is limited; however, the authors could at least strengthen the in vitro section by comparing the cytotoxic effects of the individual EDP components in the FRTL-5 cell line.
[Reply to reviewer 1, question 1] Thank you for your observation; however, we would like to clarify that, to the best of our knowledge, no published evidence supports a direct association between the EDP regimen and the development of central hypothyroidism or dyslipidemia. In contrast, both these alterations are well-known adverse effects of mitotane therapy. For this reason, our analysis focused primarily on mitotane-related comorbidities. Regarding the in vitro experiments, our study was specifically designed to explore mitotane’s direct effects on thyroid cells. Therefore, our analysis did not aim to evaluate the potential effects of the chemotherapy agents.
Question 2. The doubling time of the FRTL-5 cell line is approximately 36–38 hours, and it is generally preferable to treat cells for a duration longer than their doubling time. Why did the authors choose 24 and 48 hours as treatment durations? I suggest using 48 and 96 hours instead. Additionally, the standard deviations in the graphs are quite high, making interpretation of the results challenging. How many times were the experiments repeated? Please specify.
[Reply to reviewer 1, question 2]
Thank you for your observation, however, we would like to clarify that many papers measuring viability, secretion, signaling or gene expression in FRTL-5 use 24 and 48 hour timepoints as standard sampling times. This makes results comparable between studies and fits standard in-vitro assay schedules. 24 h is a good treatment time to detect early transcriptional, signaling or functional responses that occur within a single day and before most cells have completed a full cell cycle/doubling. Moreover, because the FRTL-5 doubling time (36–38 h) is between 24 and 48 h, a 48-hour timepoint samples the system after one full doubling period (or just beyond it), so you can observe effects that need more time to appear (protein accumulation, secreted factors, full transcriptional programs, etc.). Several studies explicitly list 48 h as an informative sampling time for functional readouts. Finally, standard in-vitro assays (like MTT) are optimized for 24–48 h windows because longer exposures can cause nutrient depletion, overconfluence, or secondary effects.
The MTT experiments were repeated independently three times, each in technical triplicate , in order to ensure reproducibility across independent culture batches. The observed variability reflects normal biological differences between independent experiments combined with technical variation, which can remain substantial even when each experiment is performed in triplicate. To strengthen future work, we plan to increase the number of independent replicates or reduce variability by further optimizing culture conditions. We expanded these points in the Methods section (Lines 134-135 and 145-147)
Question 3. To increase the relevance of the in vitro component, it would also be helpful to compare the effects of mitotane and EDP in an ACC cell line such as NCI-H295R.
[Reply to reviewer 1, question 3] We would like to clarify that the aim of our in vitro experiments was not to investigate the effects of mitotane on adrenocortical carcinoma cells, but rather to explore its direct cytotoxic impact on thyroid follicular cells, which may underlie the thyroid dysfunction observed clinically. For this reason, we used the FRTL-5 thyroid cell line, which is a well-established and physiologically relevant model for studying thyroid cell viability, hormone synthesis, and TSH responsiveness. In contrast, NCI-H295R cells are derived from adrenocortical carcinoma and can not be considered as a model to investigate thyroid-specific toxicity. Moreover, other authors explored the effect of mitotane and other cytotoxic regimen in ACC cells line, not only H295R (10.1007/s12020-014-0374-z, 10.1055/a-1105-6332, 10.1210/en.2015-1367).
Question 4. To better connect the treatments to hypothyroidism, molecular analyses are needed. For example, as the authors mention in the discussion, it would be valuable to explore the effect of mitotane on thyroid function through the measurement of thyroid hormones and thyroglobulin in the supernatant of thyroid cell cultures.
[Reply to reviewer 1, question 4] Thank you for this suggestion. We agree that measuring thyroid hormones and thyroglobulin in the culture supernatant would provide further insights. As mentioned in the discussion, this approach is planned for future studies. The current work focused primarily on the cytotoxic effects of mitotane on thyroid cells, and resources for hormone quantification were beyond its scope. We further clarified this point in the revised manuscript (Lines 362-366).
Reviewer 2 Report
Comments and Suggestions for Authors
In this study the authors found that mitotane in patients with ACC directly damages thyroid cells, through laboratory experiments showing a dose-dependent toxic effect on thyroid tissue.
Major Comments
- The manuscript describes the study as “observational cross-sectional,” yet the analyses clearly rely on longitudinal follow-up. It would greatly help readers if you could clarify whether the study is more accurately defined as a retrospective (or retrospective–prospective) cohort. Adding a simple flow diagram of patient inclusion and exclusion would further improve transparency and help readers understand the selection pathway.
- The criteria for diagnosing central hypothyroidism are briefly mentioned, but additional detail would enhance clarity. Please specify whether a single FT4/TSH pair was sufficient, or whether repeated confirmatory measurements were required. Likewise, providing the exact ESC/EAS cut-off values used for dyslipidemia—and explaining how you distinguished “pre-existing” from “mitotane-induced” dyslipidemia, particularly in borderline cases—would help clinicians apply your definitions in practice.
- TTR is an informative and clinically relevant metric. It would be very helpful if you could elaborate on how it was calculated (sampling frequency, handling of missing data, use of interpolation, etc.). You might also consider discussing whether additional exposure metrics (such as peak mitotane concentrations or cumulative exposure) were evaluated or could provide complementary insights into toxicity patterns.
- Your findings on sex-specific risks are interesting and potentially clinically meaningful. However, given the marked sex imbalance in the cohort (74% female), a broader discussion of potential confounders would be valuable. For example: were there baseline differences between men and women in ACC stage, hormonal secretion, chemotherapy exposure, or mitotane TTR that could influence these associations? Addressing these points would help contextualize the conclusions.
- The in vitro experiments nicely complement the clinical data, but providing more methodological detail (number of biological and technical replicates, controls used, and statistical tests) would strengthen confidence in the experimental findings.
In addition, there appears to be some divergence between the MTT results (IC50 ≈ 50 μM) and Incucyte® confluence behavior at the same concentration; clarification of this point would be appreciated.
Finally, the suggestion that thyroid cells might be more sensitive than pituitary cells to similar mitotane concentrations may require a more cautious wording, given the different models and experimental conditions. - The lack of association between central hypothyroidism and dyslipidemia is clearly presented; however, the small number of events and wide confidence intervals limit the strength of this conclusion. Explicitly acknowledging the limited statistical power would help avoid potential overinterpretation and guide readers toward a more cautious understanding.
Minor Comments
- You may wish to revise the study design terminology to reflect a cohort approach rather than a cross-sectional one. Also, please ensure consistency in reporting time units (some sections use months, others days).
- Only a small subset of patients had lipid measurements after mitotane discontinuation. It would be helpful to clarify that these observations are limited and not intended to be generalized.
- Please indicate whether proportional hazards assumptions were tested for the Cox models and briefly comment on the possibility of overfitting, considering the relatively small number of events.
Author Response
Reviewer #2
In this study the authors found that mitotane in patients with ACC directly damages thyroid cells, through laboratory experiments showing a dose-dependent toxic effect on thyroid tissue.
Major Comments
- The manuscript describes the study as “observational cross-sectional,” yet the analyses clearly rely on longitudinal follow-up. It would greatly help readers if you could clarify whether the study is more accurately defined as a retrospective (or retrospective–prospective) cohort. Adding a simple flow diagram of patient inclusion and exclusion would further improve transparency and help readers understand the selection pathway.
[Reply to reviewer 2, question 1] Thank you for your suggestion. The study is more accurately defined as a retrospective cohort study. We have corrected the manuscript to reflect this point (Material and Methods, Line 89). Additionally, following your suggestion, we have included a flow-chart illustrating patient inclusion and exclusion criteria. (Appendix Figure A1).
- The criteria for diagnosing central hypothyroidism are briefly mentioned, but additional detail would enhance clarity. Please specify whether a single FT4/TSH pair was sufficient, or whether repeated confirmatory measurements were required. Likewise, providing the exact ESC/EAS cut-off values used for dyslipidemia—and explaining how you distinguished “pre-existing” from “mitotane-induced” dyslipidemia, particularly in borderline cases—would help clinicians apply your definitions in practice.
[Reply to reviewer 2, question 2] Thank you for your comment which allowed us to better clarify these points. For central hypothyroidism, diagnosis required at least two repeated FT4 and TSH measurements; we specified this in Lines 110-111. For dyslipidemia, we applied the ESC/EAS 2025 cut-offs (Total Cholesterol ≥200 mg/dL, LDL ≥130 mg/dL, Triglycerides ≥150 mg/dL, HDL <40 mg/dL in men, <50 mg/dL in women). We distinguished pre-existing dyslipidemia from mitotane-induced dyslipidemia by reviewing baseline lipid profiles before mitotane initiation (Lines 114-116)
- TTR is an informative and clinically relevant metric. It would be very helpful if you could elaborate on how it was calculated (sampling frequency, handling of missing data, use of interpolation, etc.). You might also consider discussing whether additional exposure metrics (such as peak mitotane concentrations or cumulative exposure) were evaluated or could provide complementary insights into toxicity patterns.
[Reply to reviewer 2, question 3] The time in therapeutic range (TTR) was calculated following the methodology described by Puglisi S. et al., 2020 (Cancers 12(3):740), which has been previously validated in patients treated with mitotane. While we focused on TTR as the primary exposure metric, we acknowledge that additional parameters, such as peak mitotane concentrations or cumulative exposure, could provide complementary insights into toxicity patterns and may be explored in future studies.
- Your findings on sex-specific risks are interesting and potentially clinically meaningful. However, given the marked sex imbalance in the cohort (74% female), a broader discussion of potential confounders would be valuable. For example: were there baseline differences between men and women in ACC stage, hormonal secretion, chemotherapy exposure, or mitotane TTR that could influence these associations? Addressing these points would help contextualize the conclusions.
[Reply to reviewer 2, question 4] Thank you for this insightful comment. It is known from the literature that adrenocortical carcinoma shows a modest female predominance (female-to-male ratio ≈ 1.5–1.7). However, in our cohort, baseline analyses did not reveal significant differences between female and male patients regarding disease stage, hormonal secretion, thyroid/lipidic profile at diagnosis and mitotane TTR. We therefore caution that, given the small number of male patients and the known female predominance in ACC, the observed sex-specific associations should be interpreted as preliminary data and confirmed in larger studies.
- The in vitro experiments nicely complement the clinical data, but providing more methodological detail (number of biological and technical replicates, controls used, and statistical tests) would strengthen confidence in the experimental findings.
[Reply to reviewer 2, question 5]
Thank you for this comment. The MTT assay was repeated three independent times (biological replicates), with each condition performed in technical triplicate. Untreated cells (with and without TSH) were used as negative controls, and for each treatment timepoint the cell viability of the mitotane doses was compared to its corresponding untreated control. IC50 curves were generated using the log(inhibitor) vs. normalized response — variable slope model with least squares fit. More data was added in the Methods section.
- In addition, there appears to be some divergence between the MTT results (IC50 ≈ 50 μM) and Incucyte® confluence behavior at the same concentration; clarification of this point would be appreciated.
[Reply to reviewer 2, question 6]
Thank you for this significant comment. The apparent divergence arises because the two assays measure different biological parameters: the MTT assay quantifies metabolic activity based on mitochondrial reduction of tetrazolium to formazan, whereas the Incucyte® confluence analysis measures changes in cell-covered area over time, reflecting proliferation and morphology rather than metabolic rate. We decided to perform these two assays precisely to give a more complete overview of the effects of mitotane on our cellular model.
- Finally, the suggestion that thyroid cells might be more sensitive than pituitary cells to similar mitotane concentrations may require a more cautious wording, given the different models and experimental conditions.
[Reply to reviewer 2, question 7] Thank you for your comment. We rephrased Lines 355- 356 as suggested.
- The lack of association between central hypothyroidism and dyslipidemia is clearly presented; however, the small number of events and wide confidence intervals limit the strength of this conclusion. Explicitly acknowledging the limited statistical power would help avoid potential overinterpretation and guide readers toward a more cautious understanding.
[Reply to reviewer 2, question 8] Thank you for your comment. We agree that the small number of events and wide confidence intervals limit the strength of conclusions regarding the lack of association between central hypothyroidism and dyslipidemia. We have now added this point in the limitation section (Lines 431-434).
Minor Comments
- You may wish to revise the study design terminology to reflect a cohort approach rather than a cross-sectional one. Also, please ensure consistency in reporting time units (some sections use months, others days). We modified it in the text as suggested.
- Only a small subset of patients had lipid measurements after mitotane discontinuation. It would be helpful to clarify that these observations are limited and not intended to be generalized. We specified it in the Limitations section.
- Please indicate whether proportional hazards assumptions were tested for the Cox models and briefly comment on the possibility of overfitting, considering the relatively small number of events. We assessed the proportional hazards assumption and found no significant violations. Given the relatively small number of events, some degree of overfitting cannot be entirely excluded, although we limited the number of covariates and performed sensitivity analyses to mitigate this risk. We better explained this topic in the Limitation section (Lines 431-435).
Reviewer 3 Report
Comments and Suggestions for Authors
In this manuscript the authors analyzed the incidence of hypothyroidism (central) and dyslipidemia in patients with ACC treated with mitotane. They also showe mitotane’s direct toxic effects on thyroid cells.
Although these 2 side effects of mitotane treatment (central hypothyroidism and dyslipidemia ) are well known, the authors analyzed the time until occurence and the relationship with mitotane blood level and gender. They also showed that there is no clear relationship between hypothryoidism and dyslipidemia.
A few suggestions:
The Kaplan Meyer figures are represented as hypothyroidism and dyslipidemia-free survival, therefore the title of the figures should be changed accordingly. If the time to event (side-effect) were represented, the curve had to be drawn as an increasing one.
Figure 3 should be larger and with a more explicit legend, in order to be understood.
The small number or men should also be a recognized limitation of the study.
The references are adequate.
Author Response
Reviewer #3
In this manuscript the authors analyzed the incidence of hypothyroidism (central) and dyslipidemia in patients with ACC treated with mitotane. They also showed mitotane’s direct toxic effects on thyroid cells.
Although these 2 side effects of mitotane treatment (central hypothyroidism and dyslipidemia) are well known, the authors analyzed the time until occurrence and the relationship with mitotane blood level and gender. They also showed that there is no clear relationship between hypothyroidism and dyslipidemia.
A few suggestions:
The Kaplan Meyer figures are represented as hypothyroidism and dyslipidemia-free survival; therefore, the title of the figures should be changed accordingly. If the time to event (side-effect) were represented, the curve had to be drawn as an increasing one.
[Reply to Reviewer 3] Thank you for your comment. We updated the Kaplan Meier figure’s title as suggested.
Figure 3 should be larger and with a more explicit legend, in order to be understood.
[Reply to Reviewer 3] The definition of the figure is the largest available, then the size of the figure can be enlarged with the publication process (now I have increased it). However, according to your suggestion, we write an explanation with further details in the legend.
The small number of men should also be a recognized limitation of the study.
[Reply to Reviewer 3] We added this point in the Limitations paragraph.
The references are adequate.
Round 2
Reviewer 1 Report
Comments and Suggestions for Authors
Accept in present form
Reviewer 3 Report
Comments and Suggestions for Authors
No further corrections are needed. The paper may be published in its present form